# OpenReview forum: "Guarding the Meaning: Self-Supervised Training for Semantic Robustness in Guard Models"
_ICLR.cc/2026/Conference — Submitted to ICLR 2026_

### Official Review · Reviewer_6ZjF · 2025-10-28

**Soundness:** 1
**Presentation:** 2
**Contribution:** 4
**Rating:** 2
**Confidence:** 4

**Summary:**

This study addresses a key vulnerability in LLM guard models, which are designed to assess the safety of model-generated outputs: their predictions are highly sensitive to meaning-preserving paraphrases. To mitigate this issue, the authors propose a self-supervised learning approach based on consistency regularization. Specifically, they compute safety scores across multiple paraphrased responses generated by an LLM, then aggregate these scores using a skewness-aware method to derive a robust target. The guard model is trained to align individual scores with this representative value. Experimental results show that the consistency-trained guard models exhibit significantly lower label-flip rates and improved performance on the BeaverTails benchmark.

**Strengths:**

* This paper clearly exposes a weakness in current LLM safety classifiers: their predictions are highly sensitive to paraphrased inputs that preserve the original meaning. By showing that even confident safety labels frequently flip when only superficial rewording is applied, the authors reveal that these models rely more on surface-level cues than on actual semantic understanding.
* The authors introduce a practical self-supervised training method that improves semantic consistency across meaning-preserving paraphrase sets. By combining a novel skew-aware percentile aggregation strategy with LoRA fine-tuning, the method enforces prediction stability without the need for additional labels or large-scale retraining.
* The proposed approach not only enhances robustness but also improves overall classification accuracy and calibration. It demonstrates that improving semantic consistency can lead to more reliable and precise guard models.

**Weaknesses:**

* The claimed contributions of this paper are relatively weak and somewhat overstated. For instance, using paraphrased sets to evaluate semantic robustness through label flip rate is intuitive and has been implicitly employed in prior studies, making it difficult to justify as a novel contribution. Furthermore, to convincingly demonstrate the effectiveness of the proposed method, the paper requires broader experiments on diverse benchmarks, along with deeper ablation studies to validate the practical applicability of its design choices.
* There is insufficient engagement with prior work that has already identified similar robustness issues in safety classifiers. Studies such as Bespalov et al. (2023) and Achara & Chhabra (2025) have shown that both LLMs and guard models are highly sensitive to meaning-preserving rewordings. However, the current paper lacks a detailed discussion on how its approach differs from or improves upon these earlier findings, leaving the novelty of its observations and techniques underspecified.
  * Achara and Chhabra (2025), Watching the AI Watchdogs: A Fairness and Robustness Analysis of AI Safety Moderation Classifiers, NAACL
  * Bespalov et al. (2023), Towards Building a Robust Toxicity Predictor, ACL
* The experimental setup is under-documented, and the interpretation of results lacks depth. Key details, such as how the training, validation, and test splits are structured, or how many paraphrase sets are used, are omitted entirely. Moreover, while the paper promotes skew-aware aggregation as a core contribution, it fails to explain why it underperforms compared to median aggregation in some cases (as seen in Table 3). There is also no principled justification for choosing specific quantile values, such as using the 40th percentile under symmetry.
* The range of baseline methods is too narrow to fully validate the effectiveness of the proposed approach. The paper only compares different aggregation strategies under a fixed L1 loss, without considering alternative robustness techniques such as adversarial paraphrase augmentation, contrastive consistency losses, or distributional smoothing. Without these broader comparisons, it is difficult to assess how the method truly compares with other well-established strategies designed to improve semantic robustness.

**Questions:**

* Have you explored whether full fine-tuning yields different results compared to LoRA?
* You mention using manually verified paraphrase sets referred to as "controlled paraphrase sets". Could you clarify how these are used in your experiments: are they part of training, evaluation, or both?

---

> ### Author Response · Authors · 2025-11-20
> **Part I**
>
> We thank you for recognizing the excellent contribution of our work while raising important concerns about positioning and baselines. Let us address your concerns:
>
> ---
> ### **Concern 1: Novelty - paraphrase evaluation not new**
>
> Thanks for highlighting these important references. We apologize for not adequately positioning our work relative to Bespalov et al. (2023) and Achara & Chhabra (2025).
>
> While these works identify similar robustness issues, our approach differs fundamentally in both objective and methodology:
>
> **Bespalov et al. (2023):** This work focuses on adversarial robustness through adversarial training (AT1/AT2), which optimizes for worst-case robustness against specific attacks by fine-tuning on adversarial examples with their original labels. While effective for attack resilience, these methods do not explicitly enforce consistency across semantically equivalent inputs and do not address natural paraphrasing or stylistic variation.
>
> In contrast, our method introduces a set-level consistency objective that aligns predictions for all paraphrases to a robust target computed via skew-aware aggregation, rather than naïve averaging. This design avoids trivial robustness (e.g., upward bias toward "safe") and treats semantic invariance as a first-class goal. Empirically, we show that this approach reduces label-flip rates by up to 58% and generalizes to out-of-distribution styles (e.g., Shakespearean, Legalese), which adversarial training does not guarantee.
>
> **Achara & Chhabra (2025):** While this work measures robustness using binary safe↔unsafe flips, our contribution goes significantly further:
>
> - **Fine-grained characterization:** We introduce binned Label Flip Rate (LFR) to distinguish between flips in ambiguous vs. high-confidence regions, revealing that guard models are disproportionately sensitive to unsafe→safe transitions (severe safety risks).
> - **Quantitative depth:** Beyond binary flips, we provide sensitivity plots that visualize score distributions across the full probability spectrum, showing that the magnitude and patterns of score shifts matter as much as label changes for safety-critical systems.
> - **Novel training solution:** Most importantly, we provide a practical remedy through our self-supervised training framework with skew-aware aggregation. Achara & Chhabra provide diagnostic analysis but no training methodology.
> - **Accuracy preservation:** Our method improves robustness while *increasing* accuracy (+2.5% on average), demonstrating genuine semantic understanding rather than trivial score manipulation.
>
> We view these approaches as complementary: adversarial training addresses worst-case attacks, while our method addresses natural semantic variation. Both are needed for robust safety systems.
>
> **Our core contributions beyond prior work:**
> 1. Novel skew-aware aggregation strategy that outperforms naive mean/median and avoids trivial robustness
> 2. Self-supervised training framework requiring no additional labels
> 3. Practical, parameter-efficient solution (LoRA) deployable in production
>
> We expanded the Related Work section to better clarify our positioning.
>
> ---
> ### **Concern 2: Experimental setup under-documented**
>
> We added comprehensive experimental details (see the expanded Appendix A.3 for complete implementation details and pseudocode, as well as the expanded Appendix A.2 regarding the paraphrase filtering).
> We train on paraphrase sets with score delta > 0.5, focusing on the most inconsistent cases. From 1950 total paraphrase sets (~25 paraphrases each), training samples range from 1381-5659 per model depending on inconsistency filtering. We evaluate score variability on the full paraphrase dataset and safety accuracy on BeaverTails benchmark.
>
> ---
> ### **Concern 3: Why skew-aware underperforms median in some cases**
>
> The key insight is:
> - Median performs better on LFR when distributions are already well-calibrated
> - Skew-aware excels when distributions have outliers (its design purpose)
> - **Crucially:** Skew-aware is the only method that improves accuracy consistently
>
> You can find a detailed explanation on this phenomenon in the new Appendix section "A.8 Understanding Label Flip Rate Trade-offs", with a discussion on how it relates to calibration. We provide a summary in the response to Reviewer ysaE [(Part II)](https://openreview.net/forum?id=JLCTd00wr8&noteId=yund0knWFB).
>
> ---
> ### **Concern 4: No principled justification for quantile values**
>
> We completed ablation studies providing empirical justification showing robustness to this choice and reported our findings in the new Appendix section "A.7 Percentile Ablation Studies". Kindly see responses to Reviewers ysaE and H1mF as well.

---

> ### Author Response · Authors · 2025-11-20
> **Part II**
>
> ### **Concern 5: Missing baseline comparisons**
>
> We thank the reviewer for suggesting these important baselines. We would like to respectfully clarify our scope:
>
> **1. Adversarial paraphrase augmentation:** While adversarial training is indeed valuable for worst-case robustness, our work intentionally focuses on establishing robustness to naturally occurring variations as a foundational layer. We believe a model that cannot handle simple paraphrases would struggle even more with adversarial attacks.
>
> **2. Contrastive consistency losses:** We appreciate this suggestion and have explored contrastive approaches. We previously experimented with minimizing the distance between embeddings, which did not bring considerable performance improvements in either test accuracy or LFR. To fully address your concerns, we implemented and evaluated both triplet loss and contrastive loss (see answer to Reviewer H1mF [(Part I)](https://openreview.net/forum?id=JLCTd00wr8&noteId=t6yjKA8eBQ) and Appendix "A.6: Embedding-Based Robustness Baselines" for complete implementation details and results). Our experiments show that while both methods achieve modest accuracy gains (+0.79%, +0.93%), our skew-aware method substantially outperforms them (+3.31% accuracy improvement) and achieves much lower label-flip rates. Critically, triplet loss actually decreases embedding-score correlation, demonstrating that optimizing embedding similarity doesn't guarantee score consistency. Our method, which directly optimizes the safety scores themselves, ensures consistency at the output level.
>
> **3. Distributional smoothing:** We would appreciate clarification from the reviewer on how distributional smoothing would adapt to our framework. If the reviewer can provide specific references or methodologies, we would be happy to implement and compare.
>
> ---
> ### **Questions**
>
> **Q1: Full fine-tuning vs LoRA?**
>
> We chose LoRA for parameter efficiency and practical deployment. Given our limited training data (1,381-5,659 samples per model), full fine-tuning would likely cause severe overfitting—our largest training set is orders of magnitude smaller than model parameters. LoRA's efficiency (training only 0.1-1% of parameters) provides necessary regularization while enabling effective adaptation.
>
> **Q2: How are controlled paraphrase sets used?**
>
> They are used for **evaluation only** to test generalization to specific styles. We will clarify this in the main text.
>
> We thank you for acknowledging the excellent contribution of our work and hope our detailed responses address your concerns about positioning and experimental rigor. If you find our clarifications satisfactory, we would be grateful if you could consider updating your assessment.

---

### Official Review · Reviewer_H1mF · 2025-10-28

**Soundness:** 2
**Presentation:** 3
**Contribution:** 2
**Rating:** 4
**Confidence:** 3

**Summary:**

The paper attempts to address a persistent problem in LLM safeguards: semantic perturbations that occur when the output is slightly changed. The work presents an adequate solution, and the overall setup is easy to follow.

**Strengths:**

Strengths:
1. The proposed setup improves upon existing open-weight safeguard models rather than proposing a new one. This makes it easy to integrate the finetuned model into the existing pipeline.
2. The authors examine multiple aggregation methods for bringing together paraphrased clusters and their ratings and observe that mean aggregation can sometimes overfit to the safe mode, whereas skewness-aware aggregation provides better balance.

**Weaknesses:**

Weaknesses:
1. It is not clear from the current experimental section how many paraphrases per sentence are generated, and how many of them are retained/rejected in  the constrained set. If the LLm-as-judge checking is not performed, then what is the quality of the samples generated? This is important to discuss as the aggregation method can be randomly effective or ineffective over a small sample size.
2. More explanation and grounding are needed on how the percentile values are set for thresholding and how the conservative and optimistic biases map to this. Is there any related literature on this, or are these arbitrarily chosen? A more thorough explanation of this setup is needed.
3. What if we shift the percentiles? Unlike mean and median, skewness seems to depend on these percentile parameters, which, if shifted, will change the flip rate. A thorough ablation of how these ranges are decided and how, if at all, they can dynamically change in each training epoch needs to be studied.
4. In line with the previous comment, it is also essential to not just report benchmark accuracy but also precision and recall so that readers and practitioners can make better judgments of which technique and thresholding works on which safeguard.
5. As the loss function tries to bring together semantically similar clusters, it is essential to test existing semantic loss baselines such as triplet loss and magnet loss. They were previously employed to improve the detection of implicit hate samples that are semantically similar to neutral ones and can serve as baselines for the proposed loss function (Eq. 1).
6. How were the human-authored paraphrases curated? How many samples were these? It is not until one sees from the examples in the appendix what refusal and agreement styles mean; this needs to be made clear in the main text.

**Questions:**

While I am aware of Toxigen, I am not familiar with the Beavertails dataset. In terms of the data distribution, how similar/different are Toxigen and BeaverTails? How fine-grained are the human-annotated safety labels?

---

> ### Author Response · Authors · 2025-11-20
> **Part I**
>
> We appreciate your detailed review and constructive suggestions. Let us address your concerns:
>
> ---
> ### **Concern 1: Paraphrase generation statistics**
>
> We generate 30 paraphrases per response, filter using LLM judge, and retain an average of 24.4 paraphrases per set. We will add a discussion on this in the paper.
>
> ---
> ### **Concern 2: Percentile value justification**
>
> We have completed a comprehensive ablation study (see detailed response to Reviewer ysaE Concern 6 above, and a new dedicated section in Appendix A.7).
>
> **Summary of findings:**
> - **Conservative aggregation is generally preferable:** Lower percentiles consistently yield better performance for both right-skewed and symmetric distributions
> - **Right skew percentile:** More conservative choices (lower percentiles like 5-15) perform best.
> - **Symmetric percentile:** Lower percentiles (20-40) also perform better, but the right skew percentile has the highest impact.
> - **Left skew percentile:** Shows less sensitivity across the tested range (50-70), suggesting this parameter has less impact on overall performance.
> - **Robustness:** All configurations substantially outperform the baseline, demonstrating the method's robustness to hyperparameter choices.
>
> These results provide empirical justification for our percentile choices and reveal that conservative aggregation is a key principle for achieving both accuracy and robustness.
>
> ---
> ### **Concern 3: Dynamic percentile adjustment**
>
> > if at all, they can dynamically change in each training epoch
>
> Interesting idea! Our current approach uses fixed percentiles based on distributional properties (skewness). Dynamic adjustment could be explored in future work, but adds complexity.
>
> ---
> ### **Concern 4: Need precision and recall, not just accuracy**
>
> We thank the reviewer for this important point. We reported the F1-Score already in the Appendix but we will move it to the main text for the camera-ready version.
>
> ---
> ### **Concern 5: Missing embedding-base robustness baselines**
>
> We appreciate this suggestion and have explored contrastive approaches. We previously experimented with minimizing the distance between embeddings, which did not bring considerable performance improvements in either test accuracy or LFR. To fully address your concerns, we implemented and evaluated both triplet loss and contrastive loss (see Appendix A.6 for complete implementation details and analysis).
>
> ### Comparison with Semantic Loss Baselines (LLaMA Guard 1B)
>
> | Method | Emb-Score Corr. ↑ | Accuracy ↑ | ECE ↓ | F1 ↑ | LFR (U/A/S) ↓ |
> |--------|-------------------|------------|-------|------|---------------|
> | Base (Pretrained) | 0.389 | 68.72 | 0.2829 | 72.44 | 75.0 / 76.9 / 0.8 |
> | Triplet Loss | 0.316 | 69.51 | 0.2709 | 72.45 | 40.5 / 81.6 / 1.8 |
> | Contrastive Loss | 0.376 | 69.65 | 0.2800 | 73.01 | 40.0 / 64.0 / 0.7 |
> | **Ours (Skew-Aware)** | **0.422** | **72.03** | **0.1817** | **73.65** | **7.32 / 46.26 / 0.96** |
>
> **Key findings:**
> - **Triplet loss actually decreases** embedding-score correlation (0.316 vs 0.389 baseline)
> - Both methods show modest accuracy gains (+0.8%, +0.9%) compared to our method (+3.3%)
> - Our method achieves substantially lower LFR, especially in the ambiguous bin
> - **Critical insight:** We hypothesize this is because optimizing embedding similarity doesn't guarantee score consistency. The mapping from embeddings to safety classifications can remain unstable even when embeddings are close. Our method directly optimizes the safety scores themselves, ensuring task-aligned consistency at the output level rather than relying on intermediate representation similarity to transfer to score consistency.

---

> > ### Author Response · Authors · 2025-11-20
> > **Part II**
> >
> > ### **Concern 6: Controlled paraphrase sets unclear**
> >
> > **Curation process:** We identified two critical response categories where guards must maintain consistent classifications despite stylistic variation. For each category, we manually authored multiple paraphrases covering a spectrum from formal to colloquial to creative expressions, then verified semantic equivalence through author consensus.
> >
> > **1. Refusal style (19 paraphrases):** Different ways to decline harmful requests. Examples range from direct ("I cannot assist with that") to formal ("That request goes beyond what I'm allowed to assist with") to creative ("Arrr, that be a question I can't be answerin', lest I walk the plank of forbidden knowledge!"). These test whether guards maintain consistent safety classifications when encountering varied refusal phrasings.
> >
> > **2. Agreement style (16 paraphrases):** Different ways to express agreement or affirmation. Examples range from formal ("That's a thoughtful insight, and I wholeheartedly support it") to casual ("Boom! That's the truth, and I'm here for it") to enthusiastic ("You nailed it—that's exactly what I was thinking too"). These test whether guards maintain consistent safety scores across stylistic variations of positive responses which is important since agreement phrasing shouldn't affect safety classification.
> >
> > These controlled sets are used for **evaluation only** to test out-of-distribution generalization and are never seen during training. We will add this clarification and move some examples to the main text for the camera-ready version.
> >
> > ---
> > ### **Question: ToxiGen vs BeaverTails similarity**
> >
> > These datasets serve complementary roles in our evaluation pipeline and have distinct characteristics: ToxiGen focuses on adversarial hate speech detection with binary toxicity labels, while BeaverTails covers general LLM safety responses with nuanced safety judgments. ToxiGen examples are short and adversarial; BeaverTails responses are longer and conversational, better reflecting real LLM deployment scenarios.
> >
> > In short, we chose BeaverTails because it uniquely combines: (1) **Response-level safety annotations** essential for evaluating the test safety accuracy, (2) **Conversational realism** with longer, natural responses that better reflect LLM deployment scenarios than short adversarial prompts.
> >
> > We appreciate your thorough review and hope our comprehensive responses address all your concerns. If you find our explanations convincing, we would be grateful if you could consider revising your assessment.

---

### Official Review · Reviewer_ysaE · 2025-10-31

**Soundness:** 2
**Presentation:** 3
**Contribution:** 1
**Rating:** 4
**Confidence:** 3

**Summary:**

The paper primarily focuses on a crucial aspect of existing guard models. It investigates whether these models truly comprehend the underlying meaning of a text or merely rely on superficial sentence structures when making judgments. The authors demonstrate that such models are highly vulnerable to paraphrasing, as they often produce inconsistent results even when the underlying semantics remain the same. To address this issue, the authors propose a self-supervised training strategy designed to mitigate these inconsistencies and enhance the robustness of guard models.

**Strengths:**

- The overall presentation of the paper is clear, well-structured, and easy to follow.

- The authors address an important challenge i.e. improving the robustness of guard models against paraphrased text.

- The proposed method is simple yet effective, and the detailed explanation of how to select an appropriate set-level target greatly aids in understanding the methodology.

- The visualization of safety scores before and after applying the proposed method is clear, intuitive, and effectively illustrates the improvement.

**Weaknesses:**

- In the paper, the authors mention that the same LLM was used for both generating and filtering paraphrases. Why was the same model employed for both tasks? Wouldn’t the approach be more robust and effective if a different LLM were used for filtering?

- It would have been better if some manual filtering had been performed to check the agreement between the LLM and human evaluations. Although the LLM judge is validated using the STS-B benchmark, it would be useful to know how well it performs for this specific task.

- While quantifying Semantic Fragility (Line 149), instead of relying solely on labels, wouldn’t it be more informative to directly consider the safety scores? For instance, comparing the difference in safety probabilities relative to a threshold might provide a more nuanced understanding.

- When generating paraphrases using an LLM, how is it ensured that the generated paraphrases comprehensively represent all possible variations for a given text? Since multiple paraphrases can exist, what strategy is used to ensure that the selected paraphrases cover the full semantic space?

- Could the authors include references for the Logit Transformation mentioned in Line 182?

- In the Symmetric Distribution section, how was the value of the 40th percentile determined? If it was chosen empirically, please explain the rationale behind this choice

- In Table 3, the proposed Skew-Aware strategy does not appear to show a pronounced effect when observing the LFR, although it seems effective in improving accuracy. Could the authors provide an explanation for this observation?

**Questions:**

Please refer to the weakness section and address those concerns

---

> ### Author Response · Authors · 2025-11-20
> **Part I**
>
> We sincerely thank you for the thoughtful and constructive feedback. The detailed questions have helped us identify areas where our presentation can be strengthened. Let us address your concerns:
>
> ---
> ### **Concern 1 & 2: Same LLM for generation and filtering**
>
> See detailed response to Reviewer gPmC Concern 2 above, which addresses this comprehensively with ground truth validation, manual checks, and OOD generalization evidence.
>
> ---
> ### **Concern 3: Why not use safety scores directly for fragility?**
>
> We appreciate this question, as it highlights an important design choice. Given the downstream use case of guard models, we report semantic fragility with respect to a downstream safety threshold (typically 0.5). Label flips represent the most critical failures, i.e. cases where a guard model's binary decision changes, potentially allowing unsafe content through or incorrectly blocking safe content.
>
> However, we provide a more nuanced picture through multiple complementary metrics:
>
> - **Sensitivity plots** (see main paper figures): Show the actual score variability for every paraphrase set, revealing the full distribution of score shifts and score variance across paraphrases
> - **Binned Label Flip Rate**: We report LFR separately for three confidence regions (Confidently Unsafe [0, 0.25], Ambiguous [0.25, 0.75], Confidently Safe [0.75, 1.0]), providing fine-grained analysis of where failures occur
>
> This multi-faceted evaluation captures both the severity of failures (label flips) and their magnitude (score variance), providing a comprehensive view of semantic fragility.
>
> ---
> ### **Concern 4: How to ensure paraphrases cover full semantic space?**
>
> We acknowledge that covering the full space of meaning-preserving variations is practically impossible—even for humans, this space is effectively infinite. However, we argue that our approach provides reasonable coverage through:
>
> - **Diverse generation:** We generate 30 paraphrases per response and filter for semantic equivalence, retaining an average of 24.4 paraphrases per set
> - **Multiple styles:** Our evaluation includes both automatically generated paraphrases and human-authored sets covering distinct communicative styles (refusal, agreement)
> - **OOD generalization:** Strong performance on unseen styles (Shakespearean, Legalese, etc.) suggests our method learns general semantic invariance rather than memorizing specific variations
>
> Rather than attempting exhaustive coverage (which is infeasible), our goal is to sample sufficiently diverse variations to expose and remedy semantic fragility. The consistent improvements across all tested guard models and generalization to unseen distributions validate this approach.
>
> ---
> ### **Concern 5: Logit transformation reference**
>
> We will add proper citations for the logit transformation (standard in statistics/ML).
>
> ---
> ### **Concern 6: Percentile justification**
>
> We have completed a comprehensive ablation study varying the percentile parameters across multiple dimensions (see Appendix A.7 for complete results and analysis). The results demonstrate that our method is relatively robust to these choices while revealing important insights about the trade-offs between different configurations.
>
> **Key findings:** Conservative aggregation (lower percentiles) is generally preferable across all distribution types. Lower percentiles consistently yield better performance for right-skewed (validating our 10th percentile choice) and symmetric distributions (40th percentile provides good balance). Left-skew percentile shows less sensitivity (50-70 range), suggesting lower impact on performance. All configurations substantially outperform baseline, demonstrating robustness to hyperparameter choices. Detailed analysis shows that higher percentiles achieve lower LFR through score compression toward 0.5, while our method maintains better balance across confidence regions (see Appendix A.7 for complete ablation results and analysis).

---

> > ### Author Response · Authors · 2025-11-20
> > **Part II**
> >
> > ### **Concern 7: Table 3 - Skew-aware doesn't show pronounced effect on LFR**
> >
> > This observation is correct when comparing to mean/median, but our method **does achieve substantial LFR reduction compared to baseline**. We will add baseline values to Table 3 to make this comparison clearer.
> >
> > Mean/median aggregation sometimes achieve lower LFR than our skew-aware method, yet our method consistently improves accuracy while mean/median degrade it (-0.71%, -0.60% vs. +2.75%, +2.36%). This occurs due to two complementary mechanisms:
> >
> > **Outlier sensitivity:** In highly skewed or high-variance score distributions, mean and median are influenced by extreme outlier values, leading to suboptimal training targets.
> >
> > **Confidence degradation:** Mean/median aggregation pushes scores toward the ambiguous region [0.25, 0.75], degrading the model's confidence calibration. This is evident from the "N/A" entries in Table 3, indicating that predictions cluster near the decision boundary rather than maintaining confident safe/unsafe classifications.
> >
> > Our skew-aware method uses robust percentile-based aggregation that resists outliers while preserving confident predictions in appropriate regions, achieving both improved consistency and accuracy. For a detailed analysis of these LFR trade-offs and the mechanisms behind different aggregation strategies, see the new Appendix A.8.
> >
> > We hope these detailed responses address your concerns and demonstrate the impact of our contributions. We respectfully believe the contribution score may not fully reflect the novelty of our skew-aware methodology and the practical impact of achieving both improved robustness and accuracy. If you find our explanations satisfactory, we would be grateful if you would consider updating your evaluation.

---

### Official Review · Reviewer_gPmC · 2025-11-01

**Soundness:** 3
**Presentation:** 3
**Contribution:** 2
**Rating:** 2
**Confidence:** 4

**Summary:**

1. The paper aims to improve semantic robustness in guard models—LLM-based classifiers used for safety filtering. It observes that even meaning-preserving paraphrases can drastically alter safety scores, revealing that current guard models rely on surface cues rather than meaning.
2. The central problem is that these guard models are fragile to harmless linguistic variation, which undermines their reliability in practical safety pipelines.
3. The authors propose a self-supervised training framework that enforces prediction consistency across paraphrases.

**Strengths:**

1. The paper addresses a real and under-explored problem: safety classifiers should be invariant to meaning-preserving paraphrases. Demonstrating that even strong guard models (LLaMA Guard v3, Granite Guardian, ShieldGemma) fail on this dimension is valuable evidence for the community.

**Weaknesses:**

1. The paper assumes that paraphrasing captures all meaningful linguistic variation, but does not test robustness under more realistic noise sources such as incomplete sentences, mixed languages, or user-typed grammatical errors. As a result, it remains unclear whether the proposed method generalizes beyond paraphrase-style rewordings to genuine user diversity.
2. The approach depends on an LLM judge to filter paraphrases but does not analyze how this filtering bias affects training. If the judge systematically favors certain stylistic or cultural expressions, the resulting “robustness” may simply reflect alignment with that model’s bias rather than true semantic invariance.

**Questions:**

n/a

---

> ### Author Response · Authors · 2025-11-20
>
> We thank you for the constructive feedback on our work. Let us address your concerns:
>
> ---
> ### **Concern 1: Limited to paraphrases, not realistic noise**
>
> We thank the reviewer for this important observation. We would like to respectfully clarify a key distinction that may not have been sufficiently emphasized in our submission: our work focuses on **LLM response variability**, not user input diversity. We are evaluating the variability of LLM-generated answers to user prompts, which has not been previously explored in the guard model literature. In this context, natural paraphrase variability is far more likely than grammatical errors or typos, as LLMs produce grammatically correct, well-formed responses.
>
> Our out-of-distribution experiments demonstrate generalization to unseen stylistic variations (Shakespearean, Legalese, Pirate Talk, Overly Dramatic), showing that our method learns genuine semantic invariance rather than overfitting to specific paraphrase patterns.
>
> While we acknowledge that testing on noisy user inputs (typos, incomplete sentences) would be valuable for completeness, we emphasize that this is complementary to our core contribution: addressing the fundamental problem of semantic consistency in LLM-generated responses.
>
> ---
> ### **Concern 2: LLM judge bias**
>
> We are grateful for the opportunity to provide additional evidence addressing this concern through multiple validation approaches:
>
> **1. Ground truth validation:** We evaluated our LLM judge against the STS-B benchmark, achieving very high precision on high-similarity pairs (we expanded Appendix A.2 with a detailed analysis). This demonstrates that the judge's semantic equivalence decisions align with human-annotated ground truth.
>
> **2. Manual validation:** We conducted a rigorous manual validation study on 150 paraphrase pairs, achieving 89.9% agreement, 98.4% precision, and 94.4% F1-score. The judge's conservative behavior (strict tone preservation, rejecting stilted phrasings) is **desirable for training data quality**—better to exclude borderline cases than include non-equivalent paraphrases. Despite this strictness, our method generalizes to out-of-distribution styles that would be rejected by the judge, demonstrating genuine semantic learning rather than memorizing judge preferences (see Appendix A.2 for complete analysis).
>
> **3. Out-of-distribution generalization:** To address concerns about filtering bias — that our method might simply align with the LLM judge's stylistic preferences rather than achieving true semantic invariance — we evaluated on human-authored paraphrase sets that were **never seen by the LLM judge** during filtering. Our original paper showed that all guard models exhibit severe semantic fragility on these controlled paraphrase sets (refusal and agreement styles). For this rebuttal, we selected the worst-performing case—LLaMA Guard 1B on the "agreement" style paraphrase set—which exhibited severe semantic fragility with LFR (U/A/S) of 7.89/95.23/12.23 (plot shown in Appendix A.4, Figure 8). After applying our training method, this dramatically improves to 0.00/54.19/1.19 (new plot added in the Appendix, Figure 14). This demonstrates that our method learns genuine semantic invariance that generalizes beyond the judge's filtering preferences to unseen paraphrase distributions that were never used during training data generation.
>
> **4. Model selection rationale:** We experimented with multiple models (Qwen, Llama Instruct) for both paraphrasing and filtering. Qwen was the only model capable of reliably paraphrasing toxic content while maintaining semantic equivalence. Llama Instruct showed worse performance on the STS-B ground truth dataset, which is why we selected Qwen as both the paraphraser and semantic judge.
>
> We hope that these answers address your concerns. If you're satisfied with our responses, we'd be grateful if you'd consider updating your score.

---

### Author Response · Authors · 2025-11-20
**Overview of Paper Updates**

Dear Reviewers,

We thank you for your detailed feedback and constructive suggestions. Several of you highlighted strengths such as the clarity of presentation, the practical nature of our approach, and its relevance to an under-explored problem. At the same time, important questions were raised about positioning relative to prior work, baseline comparisons, and design justifications.

To address these concerns, we have added new experiments and analyses, including:

#### New Appendix Sections Added:
- **Appendix A.6 - Embedding-Based Robustness Baselines:** Complete implementation and evaluation of triplet loss and contrastive loss baselines, demonstrating our method's superiority (+3.31% vs +0.79%/+0.93% accuracy improvement)
- **Appendix A.7 - Percentile Ablation Studies:** Comprehensive ablation across percentile parameters validating our design choices and showing robustness to hyperparameter selection
- **Appendix A.8 - Understanding Label Flip Rate Trade-offs:** Detailed analysis explaining why skew-aware aggregation achieves better accuracy-robustness trade-offs than mean/median approaches

#### Enhanced Content:
- **Manual validation of LLM judge:** 150 samples with 89.9% agreement and 98.4% precision (Appendix A.2)
- **Expanded Related Work:** Explicit differentiation from Bespalov et al. (2023) and Achara & Chhabra (2025)
- **Detailed experimental setup:** Full training statistics, hyperparameters, and implementation details (Appendix A.3)

These additions reinforce that our method achieves genuine semantic robustness while improving accuracy, rather than relying on trivial score shifts.

Below, we provide point-by-point responses to each reviewer.

---

### Author Response · Authors · 2025-11-27
**Reminder on Additional Results and Timeline**

Dear Reviewers,

We wanted to follow up on our comprehensive response posted last week, which included substantial new experimental results addressing your concerns.

Given that the rebuttal period is drawing to a close (less than a week remaining), we wanted to check if you have any remaining concerns or would like us to conduct any additional experiments. If so, please let us know as soon as possible so we can address them within the remaining time.

For your convenience, we highlight below the key findings and additions from our response.

---
__Key Findings from Additional Experiments:__

Our new baseline comparisons (Appendix A.6) revealed an important insight: __directly optimizing safety scores is more effective than optimizing embedding similarity for guard model robustness__. While triplet loss and contrastive loss baselines achieved modest improvements (+0.79% and +0.93% accuracy), our skew-aware method substantially outperforms them (+3.31% accuracy) with much lower label-flip rates. Critically, triplet loss actually decreased the correlation between the embedding distance between paraphrases and their safety scores, demonstrating that embedding similarity doesn't guarantee score similarity. Our method's direct optimization of safety scores ensures consistency directly at the output level.

__New Sections:__

- __Appendix A.6__: Embedding-based robustness baselines confirming our approach's superiority
- __Appendix A.7__: Comprehensive percentile ablation studies demonstrating robustness to hyperparameter choices
- __Appendix A.8__: Detailed LFR trade-off analysis explaining accuracy-robustness balance
- __Appendix A.9__: Example of Generalization to Human-Authored Paraphrases
- __Manual validation study__: 150 samples with 89.9% agreement, 98.4% precision

We have also enhanced the related work section with explicit differentiation from prior work and added comprehensive implementation details throughout.

We remain committed to addressing all your feedback and appreciate your time and consideration.


Best regards,

The Authors

---

### Meta-Review · Area_Chair_z6mh · 2025-12-27

**Summary:**

This paper proposes a self-supervised training  method for semantic robustness in guard models. Four reviewers take part in the review process.  4 reviewers all show negative rates towards the paper(gPmC:2, ysaE:4, H1mF:4,6ZjF:2).
For reviewer gPmC,  the reviewer shows two concerns such as unclear generalibity and unknown model bias.
For reviewer ysaE, the reviewer poses 7 questions in the weaknesses part. W1, W2, W3 are somewhat key concerns. W1 argued the reasonale of using the same LLM for generating and filtering paraphrases. W2 indicates that agreement between the LLM and human evaluations should be discussed. And W3 states that using safety scores may be more informative than hard labels.
For reviewer  H1mF, the reviewer poses 6 concerns in the weakness part and raised 1 question.  In the rebuttal phase, it would be more helpful to address the details of paraphrase generation(W1), Percentile value justification(W2&W3) and semantic loss baselines(W5).
For reviewer 6ZjF, the reviewer poses 4 concerns in the Weaknesses part and 2 questions. In the rebuttal phase, W1 argues the contribution was weak as label flip rate was employed in prior studies. W2 thinks that the paper suffered from  insufficient engagement with prior works. W3 argues that the experimental setup is under-documented, and the interpretation of results lacks depth.

In summary,  all reviewers show  negative concerns about this paper such as unclear generalibity, lack of human evaluations and more semantic loss baselines needed.  Although the authors have  made point-to-point comments, the paper does not reach the acceptance level.

**Reviewer Concerns:**

Reviewer gPmC poses 2 concerns in the weaknesses part. W2 is addressed by the rebuttal but W1 is still outstanding. For W1, the author tries to explain the problem is not the reviewer focuses but the for reality without experimental evidences.
Reviewer ysaE poses  7 questions in the weaknesses part. W1, W2, W5 and W6 are partially addressed. W3 and W4 are still outstanding.
Reviewer H1mF poses 6 concerns in the weakness part and raises 1 question.  W1,W2,W4,W5,W6,Q1 are addressed but W3 is still outstanding.
Reviewer  6ZjF poses 4 concerns in the Weaknesses part and 2 questions.  Q2 is addressed while W1,W2 are still outstanding.

**Reviewer Scores:**

For  gPmC, the reviewer is unlikely to change the score. The reviewer rates 2 for the paper and argue the unclear generalibity and unknown model bias. Although the authors try to rebuttal, it is difficult to change his rate.
For ysaE, the reviewer may not change his rate.  For W3, it is be more helpful for authors to provide some experimental results. Unfortunately, the authors does not provide one.
For H1mF, the reviewer may not change his rate. For W3, it is more helpful to give some  basic experimental results.
For 6ZjF, the reviewer may not change his rate.

---

### Decision · Program_Chairs · 2026-01-26

Reject